# Regularizing Black-box Models for Improved Interpretability

**Gregory Plumb**
Carnegie Mellon University
gdplumb@andrew.cmu.edu

**Maruan Al-Shedivat**
Carnegie Mellon University
alshedivat@cs.cmu.edu

**Ángel Alexander Cabrera**
Carnegie Mellon University
cabrera@cmu.edu

**Adam Perer**
Carnegie Mellon University
adamperer@cmu.edu

**Eric Xing**
CMU, Petuum Inc
epxing@cs.cmu.edu

**Ameet Talwalkar**
CMU, Determined AI
talwalkar@cmu.edu

## Abstract

Most of the work on interpretable machine learning has focused on designing either inherently interpretable models, which typically trade-off accuracy for interpretability, or post-hoc explanation systems, whose explanation quality can be unpredictable. Our method, EXPO, is a hybridization of these approaches that regularizes a model for explanation quality at training time. Importantly, these regularizers are differentiable, model agnostic, and require no domain knowledge to define. We demonstrate that post-hoc explanations for EXPO-regularized models have better explanation quality, as measured by the common fidelity and stability metrics. We verify that improving these metrics leads to significantly more useful explanations with a user study on a realistic task.

## 1 Introduction

Complex learning-based systems are increasingly shaping our daily lives. To monitor and understand these systems, we require clear explanations of model behavior. Although model interpretability has many definitions and is often application specific [Lipton, 2016], local explanations are a popular and powerful tool [Ribeiro et al., 2016] and will be the focus of this work.

Recent techniques in interpretable machine learning range from models that are interpretable *by-design* [*e.g.*, Wang and Rudin, 2015, Caruana et al., 2015] to model-agnostic *post-hoc* systems for explaining black-box models such as ensembles and deep neural networks [*e.g.*, Ribeiro et al., 2016, Lei et al., 2016, Lundberg and Lee, 2017, Selvaraju et al., 2017, Kim et al., 2018]. Despite the variety of technical approaches, the underlying goal of these methods is to develop an interpretable predictive system that produces two outputs: a prediction and its explanation.

Both by-design and post-hoc approaches have limitations. On the one hand, by-design approaches are restricted to working with model families that are inherently interpretable, potentially at the cost of accuracy. On the other hand, post-hoc approaches applied to an arbitrary model usually offer no recourse if their explanations are not of suitable quality. Moreover, recent methods that claim to overcome this apparent trade-off between prediction accuracy and explanation quality are in fact by-design approaches that impose constraints on the model families they consider [*e.g.*, Al-Shedivat et al., 2017, Plumb et al., 2018, Alvarez-Melis and Jaakkola, 2018a].

In this work, we propose a strategy called Explanation-based Optimization (EXPO) that allows us to interpolate between these two paradigms by adding an *interpretability regularizer* to the loss function used to train the model. EXPO uses regularizers based on the *fidelity* [Ribeiro et al., 2016, Plumb et al., 2018] or *stability* [Alvarez-Melis and Jaakkola, 2018a] metrics. See Section 2 for definitions.

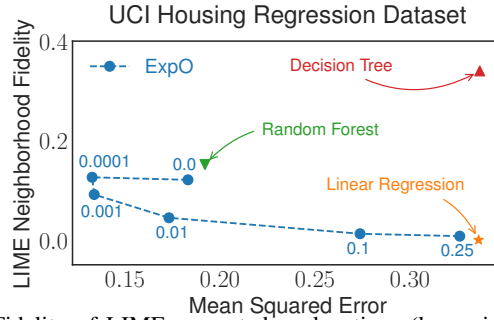

**Figure 1:** Neighborhood Fidelity of LIME-generated explanations (lower is better) vs. predictive Mean Squared Error of several models trained on the UCI 'housing' regression dataset. The values in blue denote the regularization weight of EXPO. One of the key contributions of EXPO is allowing us to pick where we are along the accuracy-interpretability curve for a black-box model.

Unlike by-design approaches, EXPO places no explicit constraints on the model family because its regularizers are differentiable and model agnostic. Unlike post-hoc approaches, EXPO allows us to control the relative importance of predictive accuracy and explanation quality. In Figure 1, we see an example of how EXPO allows us to interpolate between these paradigms and overcome their respective weaknesses.

Although fidelity and stability are standard proxy metrics, they are only indirect measurements of the usefulness of an explanation. To more rigorously test the usefulness of EXPO, we additionally devise a more realistic evaluation task where humans are asked to use explanations to change a model's prediction. Notably, our user study falls under the category of Human-Grounded Metric evaluations as defined by Doshi-Velez and Kim [2017].

The main contributions of our work are as follows:

1. **Interpretability regularizer.** We introduce, EXPO-FIDELITY, a *differentiable* and *model agnostic* regularizer that requires *no domain knowledge* to define. It approximates the fidelity metric on the training points in order to improve the quality of post-hoc explanations of the model.

2. **Empirical results.** We compare models trained with and without EXPO on a variety of regression and classification tasks.[1] Empirically, EXPO slightly improves test accuracy and significantly improves explanation quality on test points, producing at least a 25% improvement in terms of explanation fidelity. This separates it from many other methods which trade-off between predictive accuracy and explanation quality. These results also demonstrate that EXPO's effects *generalize* from the training data to unseen points.

3. **User study.** To more directly test the usefulness of EXPO, we run a user study where participants complete a simplified version of a realistic task. Quantitatively, EXPO makes it easier for users to complete the task and, qualitatively, they prefer using the EXPO-regularized model. This is additional validation that the fidelity and stability metrics are useful proxies for interpretability.

## 2 Background and Related Work

Consider a supervised learning problem where the goal is to learn a model, $f : \mathcal{X} \mapsto \mathcal{Y}$, $f \in \mathcal{F}$, that maps input feature vectors, $x \in \mathcal{X}$, to targets, $y \in \mathcal{Y}$, trained with data, $\{x_i, y_i\}_{i=1}^N$. If $\mathcal{F}$ is complex, we can understand the behavior of $f$ in some neighborhood, $N_x \in \mathcal{P}[\mathcal{X}]$ where $\mathcal{P}[\mathcal{X}]$ is the space of probability distributions over $\mathcal{X}$, by generating a *local explanation*.

We denote systems that produce local explanations (*i.e.*, *explainers*) as $e : \mathcal{X} \times \mathcal{F} \mapsto \mathcal{E}$, where $\mathcal{E}$ is the set of possible explanations. The choice of $\mathcal{E}$ generally depends on whether or not $\mathcal{X}$ consists of semantic features. We call features *semantic* if users can reason about them and understand what changes in their values mean (*e.g.*, a person's income or the concentration of a chemical). Non-semantic features lack an inherent interpretation, with images as a canonical example. We primarily focus on semantic features but we briefly consider non-semantic features in Appendix A.8.

We next state the goal of local explanations for semantic features, define fidelity and stability (the metrics most commonly used to quantitatively evaluate the quality of these explanations), and briefly summarize the post-hoc explainers whose explanations we will use for evaluation.

**Goal of Approximation-based Local Explanations.** For semantic features, we focus on local explanations that try to predict how the model's output would change if the input were perturbed such as LIME [Ribeiro et al., 2016] and MAPLE [Plumb et al., 2018]. Thus, we can define the output space of the explainer as $\mathcal{E}_s := \{g \in \mathcal{G} \mid g : \mathcal{X} \mapsto \mathcal{Y}\}$, where $\mathcal{G}$ is a class of interpretable functions. As is common, we assume that $\mathcal{G}$ is the set of linear functions.

**Fidelity metric.** For semantic features, a natural choice for evaluation is to measure how accurately $g$ models $f$ in a neighborhood $N_x$ [Ribeiro et al., 2016, Plumb et al., 2018]:

$$F(f, g, N_x) := \mathbb{E}_{x' \sim N_x}[(g(x') - f(x'))^2], \qquad (1)$$

which we refer to as the *neighborhood-fidelity* (NF) metric. This metric is sometimes evaluated with $N_x$ as a point mass on $x$ and we call this version the *point-fidelity* (PF) metric.[2] Intuitively, an explanation with good fidelity (lower is better) accurately conveys which patterns the model used to make this prediction (*i.e.*, how each feature influences the model's prediction around this point).

**Stability metric.** In addition to fidelity, we are interested in the degree to which the explanation changes between points in $N_x$, which we measure using the *stability metric* [Alvarez-Melis and Jaakkola, 2018a]:

$$S(f, e, N_x) := \mathbb{E}_{x' \sim N_x}[||e(x, f) - e(x', f)||_2^2] \qquad (2)$$

Intuitively, more stable explanations (lower is better) tend to be more trustworthy [Alvarez-Melis and Jaakkola, 2018a,b, Ghorbani et al., 2017].

**Post-hoc explainers.** Various explainers have been proposed to generate local explanations of the form $g : \mathcal{X} \mapsto \mathcal{Y}$. In particular, LIME [Ribeiro et al., 2016], one of the most popular post-hoc explanation systems, solves the following optimization problem:

$$e(x, f) := \arg\min_{g \in \mathcal{E}_s} F(f, g, N_x) + \Omega(g), \qquad (3)$$

where $\Omega(g)$ stands for an additive regularizer that encourages certain desirable properties of the explanations (*e.g.*, sparsity). Along with LIME, we consider another explanation system, MAPLE [Plumb et al., 2018]. Unlike LIME, its neighborhoods are learned from the data using a tree ensemble rather than specified as a parameter.

## 2.1 Related Methods

There are three methods that consider problems conceptually similar to EXPO: Functional Transparency for Structured Data (FTSD) [Lee et al., 2018, 2019], Self-Explaining Neural Networks (SENN) [Alvarez-Melis and Jaakkola, 2018a], and Right for the Right Reasons (RRR) [Ross et al., 2017]. In this section, we contrast EXPO to these methods along several dimensions: whether the proposed regularizers are (i) differentiable, (ii) model agnostic, and (iii) require domain knowledge; whether the (iv) goal is to change an explanation's quality (*e.g.*, fidelity or stability) or its content (*e.g.*, how each feature is used); and whether the expected explainer is (v) neighborhood-based (*e.g.*, LIME or MAPLE) or gradient-based (*e.g.*, Saliency Maps [Simonyan et al., 2013]). These comparisons are summarized in Table 1, and we elaborate further on them in the following paragraphs.

FTSD has a very similar high-level objective to EXPO: it regularizes black-box models to be more locally interpretable. However, it focuses on graph and time-series data and is not well-defined for general tabular data. Consequently, our technical approaches are distinct. First, FTSD's local neighborhood and regularizer definitions are different from ours. For graph data, FTSD aims to understand what the model would predict if the graph itself were modified. Although this is the same type of local interpretability considered by EXPO, FTSD requires domain knowledge to define $N_x$ in order to consider plausible variations of the input graph. These definitions do not apply to general tabular data. For time-series data, FTSD aims to understand what the model will predict for the next

**Table 1:** A breakdown of how ExPO compares to existing methods. Note that ExPO is the only method that is differentiable and model agnostic that does not require domain knowledge.

| Method | Differentiable | Model Agnostic | Domain Knowledge | Goal | Type |
|--------|----------------|----------------|------------------|------|------|
| ExpO | Yes | Yes | No | Quality | Neighborhood |
| FTSD | No | Yes | Sometimes | Quality | Neighborhood |
| SENN | Yes | No | No | Quality | Gradient |
| RRR | Yes | Yes | Yes | Content | Gradient |

point in the series and defines $N_x$ as a windowed-slice of the series to do so. This has no analogue for general tabular data and is thus entirely distinct from ExPO. Second, FTSD's regularizers are non-differentiable, and thus it requires a more complex, less efficient bi-level optimization scheme to train the model.

SENN is a by-design approach that optimizes the model to produce stable explanations. For both its regularizer and its explainer, it assumes that the model has a specific structure. In Appendix A.1, we show empirically that ExPO is a more flexible solution than SENN via two results. First, we show that we can train a significantly more accurate model with ExPO-Fidelity than with SENN; although the ExPO-regularized model is slightly less interpretable. Second, if we increase the weight of the ExPO-Fidelity regularizer so that the resulting model is as accurate as SENN, we show that the ExPO-regularized model is much more interpretable.

RRR also regularizes a black-box model with a regularizer that involves a model's explanations. However, it is motivated by a fundamentally different goal and necessarily relies on extensive domain knowledge. Instead of focusing on explanation quality, RRR aims to restrict what features are used by the model itself, which will be reflected in the model's explanations. This relies on a user's domain knowledge to specify sets of good or bad features. In a similar vein to RRR, there are a variety of methods that aim to change the model in order to align the content of its explanations with some kind of domain knowledge [Du et al., 2019, Weinberger et al., 2019, Rieger et al., 2019]. As a result, these works are orthogonal approaches to ExPO.

Finally, we briefly mention two additional lines of work that are also in some sense related to ExPO. First, Qin et al. [2019] proposed a method for local linearization in the context of adversarial robustness. Because its regularizer is based on the model's gradient, it will have the same issues with flexibility, fidelity, and stability discussed in Appendix A.2. Second, there is a line of work that regularizes black-box models to be easier to approximate by decision trees. Wu et al. [2018] does this from a global perspective while Wu et al. [2019] uses domain knowledge to divide the input space into several regions. However, small decision trees are difficult to explain locally by explainer's such as LIME (as seen in Figure 1) and so these methods do not solve the same problem as ExPO.

## 2.2 Connections to Function Approximations and Complexity

The goal of this section is to intuitively connect local linear explanations and neighborhood fidelity with classical notions of function approximation and complexity/smoothness, while also highlighting key differences in the context of local interpretability. First, neighborhood-based local linear explanations and first-order Taylor approximations both aim to use linear functions to locally approximate $f$. However, the Taylor approximation is strictly a function of $f$ and $x$ and cannot be adjusted to different neighborhood scales for $N_x$, which can lead to poor fidelity and stability. Second, Neighborhood Fidelity (NF), the Lipschitz Constant (LC), and Total Variation (TV) all approximately measure the smoothness of $f$ across $N_x$. However, a large LC or TV does not necessarily indicate that $f$ is difficult to explain across $N_x$ (e.g., consider a linear model with large coefficients which has a near zero NF but has a large LC/TV). Instead, local interpretability is more closely related to the LC or TV of the part of $f$ that cannot be explained by $e(x, f)$ across $N_x$. Additionally, we empirically show that standard $l_1$ or $l_2$ regularization techniques do not influence model interpretability. Examples and details for all of these observations are in Appendix A.2.

---

**Algorithm 1** Neighborhood-fidelity regularizer

**input** $f_\theta, x, N_x^{\mathrm{reg}}, m$
1: Sample points: $x_1', \ldots, x_m' \sim N_x^{\mathrm{reg}}$
2: Compute predictions: $\hat{y}_j(\theta) = f_\theta(x_j')$ for $j = 1, \ldots, m$
3: Produce a local linear explanation: $\beta_x(\theta) = \arg\min_\beta \sum_{j=1}^m (\hat{y}_j(\theta) - \beta^\top x_j')^2$
**output** $\frac{1}{m} \sum_{j=1}^m (\hat{y}_j(\theta) - \beta_x(\theta)^\top x_j')^2$

---

## 3 Explanation-based Optimization

Recall that the main limitation of using post-hoc explainers on arbitrary models is that their explanation quality can be unpredictable. To address this limitation, we define regularizers that can be added to the loss function and used to train an arbitrary model. This allows us to control for explanation quality without making explicit constraints on the model family in the way that by-design approaches do. Specifically, we want to solve the following optimization problem:

$$\hat{f} := \arg\min_{f \in \mathcal{F}} \frac{1}{N} \sum_{i=1}^N (\mathcal{L}(f, x_i, y_i) + \gamma \mathcal{R}(f, N_{x_i}^{\mathrm{reg}})) \tag{4}$$

where $\mathcal{L}(f, x_i, y_i)$ is a standard predictive loss (*e.g.*, squared error for regression or cross-entropy for classification), $\mathcal{R}(f, N_{x_i}^{\mathrm{reg}})$ is a regularizer that encourages $f$ to be interpretable in the neighborhood of $x_i$, and $\gamma > 0$ controls the regularization strength. Because our regularizers are differentiable, we can solve Equation 4 using any standard gradient-based algorithm; we use SGD with Adam [Kingma and Ba, 2014].

We define $\mathcal{R}(f, N_x^{\mathrm{reg}})$ based on either neighborhood-fidelity, Eq. (1), or stability, Eq. (2). In order to compute these metrics exactly, we would need to run $e$; this may be non-differentiable or too computationally expensive to use as a regularizer. As a result, EXPO consists of two main approximations to these metrics: EXPO-FIDELITY and EXPO-STABILITY. EXPO-FIDELITY approximates $e$ using a local linear model fit on points sampled from $N_x^{reg}$ (Algorithm 1). Note that it is simple to modify this algorithm to regularize for the fidelity of a *sparse* explanation. EXPO-STABILITY encourages the model to not vary too much across $N_x^{reg}$ and is detailed in Appendix A.8.

**Computational cost.** The overhead of using EXPO-FIDELITY comes from using Algorithm 1 to calculate the additional loss term and then differentiating through it at each iteration. If $x$ is $d$-dimensional and we sample $m$ points from $N_x^{reg}$, this has a complexity of $O(d^3 + d^2 m)$ plus the cost to evaluate $f$ on $m$ points. Note that $m$ must be at least $d$ in order for this loss to be non-zero, thus making the complexity $\Omega(d^3)$. Consequently, we introduce a randomized version of Algorithm 1, EXPO-1D-FIDELITY, that randomly selects one dimension of $x$ to perturb according to $N_x^{reg}$ and penalizes the error of a local linear model along that dimension. This variation has a complexity of $O(m)$ plus the cost to evaluate $f$ on $m$ points, and allows us to use a smaller $m$.[3]

## 4 Experimental Results

In our main experiments, we demonstrate the effectiveness of EXPO-FIDELITY and EXPO-1D-FIDELITY on datasets with semantic features using seven regression problems from the UCI collection [Dheeru and Karra Taniskidou, 2017], the 'MSD' dataset[4], and 'Support2' which is an in-hospital mortality classification problem[5]. Dataset statistics are in Table 2.

We found that EXPO-regularized models are more interpretable than normally trained models because post-hoc explainers produce quantitatively better explanations for them; further, they are often more accurate. Additionally, we qualitatively demonstrate that post-hoc explanations of EXPO-regularized

**Table 2: Left.** Statistics of the datasets. **Right.** An example of LIME's explanation for a normally trained model ("None") and an EXPO-regularized model. Because these are linear explanations, each value can be interpreted as an estimate of how much the model's prediction would change if that feature's value were increased by one. Because the explanation for the EXPO-regularized model is sparser, it is easier to understand and, because it has better fidelity, these estimates are more accurate.

| Dataset | # samples | # dims |
|---|---|---|
| autompgs | 392 | 7 |
| communities | 1993 | 102 |
| day | 731 | 14 |
| housing | 506 | 11 |
| music | 1059 | 69 |
| winequality-red | 1599 | 11 |
| MSD | 515345 | 90 |
| SUPPORT2 | 9104 | 51 |

| | Data | Explanation | |
|---|---|---|---|
| Feature | Value | None | ExpO |
| CRIM | 2.5 | -0.1 | 0.0 |
| INDUS | 1.0 | 0.1 | 0.0 |
| NOX | 0.9 | -0.2 | -0.2 |
| RM | 1.4 | 0.2 | 0.2 |
| AGE | 1.0 | -0.1 | 0.0 |
| DIS | -1.2 | -0.4 | -0.2 |
| RAD | 1.6 | 0.2 | 0.2 |
| TAX | 1.5 | -0.3 | -0.1 |
| PTRATIO | 0.8 | -0.1 | -0.1 |
| B | 0.4 | 0.1 | 0.0 |
| LSTAT | 0.1 | -0.3 | -0.5 |

models tend to be simpler. In Appendix A.8, we demonstrate the effectiveness of EXPO-STABILITY for creating Saliency Maps [Simonyan et al., 2013] on MNIST [LeCun, 1998].

**Experimental setup.** We compare EXPO-regularized models to normally trained models (labeled "None"). We report model accuracy and three interpretability metrics: Point-Fidelity (PF), Neighborhood-Fidelity (NF), and Stability (S). The interpretability metrics are evaluated for two black-box explanation systems: LIME and MAPLE. For example, the "MAPLE-PF" label corresponds to the Point-Fidelity Metric for explanations produced by MAPLE. All of these metrics are calculated on test data, which enables us to evaluate whether optimizing for explanation fidelity on the training data generalizes to unseen points.

All of the inputs to the model are standardized to have mean zero and variance one (including the response variable for regression problems). The network architectures and hyper-parameters are chosen using a grid search; for more details see Appendix A.3. For the final results, we set $N_x$ to be $\mathcal{N}(x, \sigma)$ with $\sigma = 0.1$ and $N_x^{reg}$ to be $\mathcal{N}(x, \sigma)$ with $\sigma = 0.5$. In Appendix A.4, we discuss how we chose those distributions.

**Regression experiments.** Table 3 shows the effects of EXPO-FIDELITY and EXPO-1D-FIDELITY on model accuracy and interpretability. EXPO-FIDELITY frequently improves the interpretability metrics by over 50%; the smallest improvements are around 25%. Further, it lowers the prediction error on the 'communities', 'day', and 'MSD' datasets, while achieving similar accuracy on the rest. EXPO-1D-FIDELITY also significantly improves the interpretability metrics, although on average to a lesser extent than EXPO-FIDELITY does, and it has no significant effect on accuracy on average.

**A qualitative example on the UCI 'housing' dataset.** After sampling a random point $x$, we use LIME to generate an explanation at $x$ for a normally trained model and an EXPO-regularized model. Table 2 shows the example we discuss next. Quantitatively, training the model with EXPO-1D-FIDELITY decreases the LIME-NF metric from 1.15 to 0.02 (*i.e.*, EXPO produces a model that is more accurately approximated by the explanation around $x$). Further, the resulting explanation also has fewer non-zero coefficients (after rounding), and hence it is simpler because the effect is attributed to fewer features. More examples, that show similar patterns, are in Appendix A.5.

**Medical classification experiment.** We use the 'support2' dataset to predict in-hospital mortality. Since the output layer of our models is the softmax over logits for two classes, we run each explainer on each of the logits. We observe that EXPO-FIDELITY had no effect on accuracy and improved the interpretability metrics by 50% or more, while EXPO-1D-FIDELITY slightly decreased accuracy and improved the interpretability metrics by at least 25%. See Table 9 in Appendix A.6 for details.

# 5 User Study

The previous section compared EXPO-regularized models to normally trained models through quantitative metrics such as model accuracy and post-hoc explanation fidelity and stability on held-out test data. Doshi-Velez and Kim [2017] describe these metrics as Functionally-Grounded Evaluations, which are useful *proxies* for more direct applications of interpretability. To more directly

**Table 3:** Normally trained models ("None") vs. the same models trained with EXPO-FIDELITY or EXPO-1D-FIDELITY on the regression datasets. Results are shown across 20 trials (with the standard error in parenthesis). Statistically significant differences ($p = 0.05$, t-test) between FIDELITY and None are in bold and between 1D-FIDELITY and None are underlined. Because MAPLE is slow on 'MSD', we evaluate interpretability using LIME on 1000 test points.

| Metric | Regularizer | autompgs | communities | day† ($10^{-3}$) | housing | music | winequality.red | MSD |
|---|---|---|---|---|---|---|---|---|
| **MSE** | None | 0.14 (0.03) | 0.49 (0.05) | 1.000 (0.300) | 0.14 (0.05) | 0.72 (0.09) | 0.65 (0.06) | 0.583 (0.018) |
| | FIDELITY | 0.13 (0.02) | **0.46 (0.03)** | **0.002 (0.002)** | 0.15 (0.05) | 0.67 (0.09) | 0.64 (0.06) | **0.557 (0.0162)** |
| | 1D-FIDELITY | 0.13 (0.02) | 0.55 (0.04) | 5.800 (8.800) | 0.15 (0.07) | 0.74 (0.07) | 0.66 (0.06) | 0.548 (0.0154) |
| **LIME-PF** | None | 0.040 (0.011) | 0.100 (0.013) | 1.200 (0.370) | 0.14 (0.036) | 0.110 (0.037) | 0.0330 (0.0130) | 0.116 (0.0181) |
| | FIDELITY | **0.011 (0.003)** | **0.080 (0.007)** | **0.041 (0.007)** | **0.057 (0.017)** | **0.066 (0.011)** | **0.0025 (0.0006)** | **0.0293 (0.00709)** |
| | 1D-FIDELITY | 0.029 (0.007) | 0.079 (0.026) | 0.980 (0.380) | 0.064 (0.017) | 0.080 (0.039) | 0.0029 (0.0011) | 0.057 (0.0079) |
| **LIME-NF** | None | 0.041 (0.012) | 0.110 (0.012) | 1.20 (0.36) | 0.140 (0.037) | 0.112 (0.037) | 0.0330 (0.0140) | 0.117 (0.0178) |
| | FIDELITY | **0.011 (0.003)** | **0.079 (0.007)** | **0.04 (0.07)** | **0.057 (0.018)** | **0.066 (0.011)** | **0.0025 (0.0006)** | **0.029 (0.007)** |
| | 1D-FIDELITY | 0.029 (0.007) | 0.080 (0.027) | 1.00 (0.39) | 0.064 (0.017) | 0.080 (0.039) | 0.0029 (0.0011) | 0.0575 (0.0079) |
| **LIME-S** | None | 0.0011 (0.0006) | 0.022 (0.003) | 0.150 (0.021) | 0.0047 (0.0012) | 0.0110 (0.0046) | 0.00130 (0.00057) | 0.0368 (0.00759) |
| | FIDELITY | **0.0001 (0.0003)** | **0.005 (0.001)** | **0.004 (0.004)** | **0.0012 (0.0002)** | **0.0023 (0.0004)** | **0.00007 (0.00002)** | **0.00171 (0.00034)** |
| | 1D-FIDELITY | 0.0008 (0.0003) | 0.018 (0.008) | 0.100 (0.047) | 0.0025 (0.0007) | 0.0084 (0.0052) | 0.00016 (0.00005) | 0.0125 (0.00291) |
| **MAPLE-PF** | None | 0.0160 (0.0088) | 0.16 (0.02) | 1.0000 (0.3000) | 0.057 (0.024) | 0.17 (0.06) | 0.0130 (0.0078) | — |
| | FIDELITY | **0.0014 (0.0006)** | **0.13 (0.01)** | **0.0002 (0.0003)** | **0.028 (0.013)** | 0.14 (0.03) | **0.0027 (0.0010)** | — |
| | 1D-FIDELITY | 0.0076 (0.0038) | 0.092 (0.03) | 0.7600 (0.3000) | 0.027 (0.012) | 0.13 (0.05) | 0.0016 (0.0007) | — |
| **MAPLE-NF** | None | 0.0180 (0.0097) | 0.31 (0.04) | 1.2000 (0.3200) | 0.066 (0.024) | 0.18 (0.07) | 0.0130 (0.0079) | — |
| | FIDELITY | **0.0015 (0.0006)** | **0.24 (0.05)** | **0.0003 (0.0004)** | **0.033 (0.014)** | **0.14 (0.03)** | **0.0028 (0.0010)** | — |
| | 1D-FIDELITY | 0.0084 (0.0040) | 0.16 (0.05) | 0.9400 (0.3600) | 0.032 (0.013) | 0.14 (0.06) | 0.0017 (0.0008) | — |
| **MAPLE-S** | None | 0.0150 (0.0099) | 1.2 (0.2) | 0.0003 (0.0008) | 0.18 (0.14) | 0.08 (0.06) | 0.0043 (0.0020) | — |
| | FIDELITY | **0.0017 (0.0005)** | **0.8 (0.4)** | 0.0004 (0.0004) | **0.10 (0.08)** | **0.05 (0.02)** | **0.0009 (0.0004)** | — |
| | 1D-FIDELITY | 0.0077 (0.0051) | 0.6 (0.2) | 1.2000 (0.6600) | 0.09 (0.06) | 0.04 (0.02) | 0.0004 (0.0002) | — |

†The relationship between inputs and targets on the 'day' dataset is very close to linear and hence all errors are orders of magnitude smaller than across other datasets.

measure the usefulness of EXPO, we conduct a user study to obtain Human-Grounded Metrics [Doshi-Velez and Kim, 2017], where real people solve a simplified task.

In summary, the results of our user study show that the participants had an easier time completing this task with the EXPO-regularized model and found the explanations for that model more useful. See Table 4 and Figure 8 in Appendix A.7 for details. Not only is this additional evidence that the fidelity and stability metrics are good proxies for interpretability, but it also shows that they remain so after we directly optimize for them. Next, we describe the high-level task, explain the design choices of our study, and present its quantitative and qualitative results.

**Defining the task.** One of the common proposed use cases for local explanations is as follows. A user is dissatisfied with the prediction that a model has made about them, so they request an explanation for that prediction. Then, they use that explanation to determine what changes they should make in order to receive the desired outcome in the future. We propose a similar task on the UCI 'housing' regression dataset where the goal is to increase the model's prediction by a fixed amount.

We simplify the task in three ways. First, we assume that all changes are equally practical to make; this eliminates the need for any prior domain knowledge. Second, we restrict participants to changing a single feature at a time by a fixed amount; this reduces the complexity of the required mental math. Third, we allow participants to iteratively modify the features while getting new explanations at each point; this provides a natural quantitative measure of explanation usefulness, via the number of changes required to complete the task.

**Design Decisions**. Figure 2 shows a snapshot of the interface we provide to participants. Additionally, we provide a demo video of the user study in the Github repository. Next, we describe several key design aspects of our user study, all motivated by the underlying goal of isolating the effect of EXPO.

1. **Side-by-side conditions**. We present the two conditions side-by-side with the same initial point. This design choice allows the participants to directly compare the two conditions and allows us to gather their preferences between the conditions. It also controls for the fact that a model may be more difficult to explain for some $x$ than for others. Notably, while both conditions have the same initial point, each condition is modified independently. With the conditions shown side-by-side, it may be possible for a participant to use the information gained by solving one condition first to help solve the other condition. To prevent this from biasing our aggregated results, we randomize, on a per-participant basis, which model is shown as Condition A.

2. **Abstracted feature names and magnitudes**. In the explanations shown to users, we abstract feature names and only show the magnitude of each feature's expected impact. Feature names are abstracted in order to prevent participants from using prior knowledge to inform their decisions. Moreover, by only showing feature magnitudes, we eliminate double negatives (*e.g.*, decreasing a

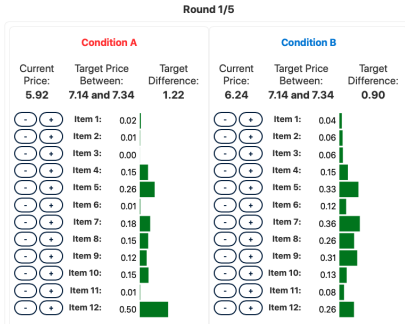

**Figure 2:** An example of the interface participants were given. In this example, the participant has taken one step for Condition A and no steps for Condition B. While the participant selected '+' for Item 7 in Condition A, the change had the opposite effect and decreased the price because of the explanation's low fidelity.

**Table 4:** The results of the user study. Participants took significantly fewer steps to complete the task using the EXPO-regularized model, and thought that the post-hoc explanations for it were both more useful for completing the task and better matched their expectations of how the model would change.

| Condition | Steps | Usefulness | Expectation |
|---|---|---|---|
| **None** | 11.45 | 11 | 11 |
| **ExpO** | 8.00 | 28 | 26 |
| **No Preference** | *N.A.* | 15 | 17 |

feature with a negative effect on the prediction should increase the prediction), thus simplifying participants' required mental computations. In other words, we simplify the interface so that the '+' button is expected to increase the prediction by the amount shown in the explanation regardless of explanation's (hidden) sign.

3. **Learning Effects**. To minimize long-term learning (*e.g.*, to avoid learning general patterns such as 'Item 7's explanation is generally unreliable.'), participants are limited to completing a single experiment consisting of five recorded rounds. In Figure 7 from Appendix A.7, we show that the distribution of the number of steps it takes to complete each round across participants does not change substantially. This result indicates that learning effects were not significant.

4. **Algorithmic Agent**. Although the study is designed to isolate the effect EXPO has on the usefulness of the explanations, entirely isolating its effect is impossible with human participants. Consequently, we also evaluate the performance of an algorithmic agent that uses a simple heuristic that relies only on the explanations. See Appendix A.7 for details.

**Collection Procedure.** We collect the following information using Amazon Mechanical Turk:

1. **Quantitative.** We measure how many steps (*i.e.*, feature changes) it takes each participant to reach the target price range for each round and each condition.

2. **Qualitative Preferences.** We ask which condition's explanations are more *useful* for completing the task and better match their *expectation* of how the price should change.[6]

3. **Free Response Feedback.** We ask why participants preferred one condition over the other.

**Data Cleaning.** Most participants complete each round in between 5 and 20 steps. However, there are a small number of rounds that take 100's of steps to complete, which we hypothesize to be random clicking. See Figure 6 in Appendix A.7 for the exact distribution. As a result, we remove any participant who has a round that is in the top 1% for number of steps taken (60 participants with 5 rounds per participant and 2 conditions per round gives us 600 observed rounds). This leaves us with a total of 54 of the original 60 participants.

**Results.** Table 4 shows that the EXPO-regularized model has both quantitatively and qualitatively more useful explanations. Quantitatively, participants take 8.00 steps on average with the EXPO-regularized model, compared to 11.45 steps for the normally trained model ($p = 0.001$, t-test). The participants report that the explanations for the EXPO-regularized model are both more useful for completing the task ($p = 0.012$, chi-squared test) and better aligned with their expectation of how the model would change ($p = 0.042$, chi-squared test). Figure 8 in Appendix A.7 shows that the algorithmic agent also finds the task easier to complete with the EXPO-regularized model. This agent relies solely on the information in the explanations and thus is additional validation of these results.

**Participant Feedback.** Most participants who prefer the EXPO-regularized model focus on how well the explanation's predicted change matches the actual change. For example, one participant says "It [EXPO ] seemed to do what I expected more often" and another notes that "In Condition A [None] the predictions seemed completely unrelated to how the price actually changed." Although some participants who prefer the normally trained model cite similar reasons, most focus on how quickly they can reach the goal rather than the quality of the explanation. For example, one participant plainly states that they prefer the normally trained model because "The higher the value the easier to hit [the] goal"; another participant similarly explains that "It made the task easier to achieve." These participants likely benefited from the randomness of the low-fidelity explanations of the normally trained model, which can jump unexpectedly into the target range.

## 6 Conclusion

In this work, we regularize black-box models to be more interpretable with respect to the fidelity and stability metrics for local explanations. We compare EXPO-FIDELITY, a model agnostic and differentiable regularizer that requires no domain knowledge to define, to classical approaches for function approximation and smoothing. Next, we demonstrate that EXPO-FIDELITY slightly improves model accuracy and significantly improves the interpretability metrics across a variety of problem settings and explainers on unseen test data. Finally, we run a user study demonstrating that an improvement in fidelity and stability improves the usefulness of the model's explanations.

## 7 Broader Impact

Our user study plan has been approved by the IRB to minimize any potential risk to the participants, and the datasets used in this work are unlikely to contain sensitive information because they are public and well-studied. Within the Machine Learning community, we hope that EXPO will help encourage Interpretable Machine Learning research to adopt a more quantitative approach, both in the form of proxy evaluations and user studies. For broader societal impact, the increased interpretability of models trained with EXPO should be a significant benefit. However, EXPO does not address some issues with local explanations such as their susceptibility to adversarial attack or their potential to artificially inflate people's trust in the model.

## Acknowledgments

This work was supported in part by DARPA FA875017C0141, the National Science Foundation grants IIS1705121 and IIS1838017, an Okawa Grant, a Google Faculty Award, an Amazon Web Services Award, a JP Morgan A.I. Research Faculty Award, and a Carnegie Bosch Institute Research Award. Any opinions, findings and conclusions or recommendations expressed in this material are those of the author(s) and do not necessarily reflect the views of DARPA, the National Science Foundation, or any other funding agency. We would also like to thank Liam Li, Misha Khodak, Joon Kim, Jeremy Cohen, Jeffrey Li, Lucio Dery, Nick Roberts, and Valerie Chen for their helpful feedback.

## Footnotes

[1]https://github.com/GDPlumb/ExpO

[2]Although Plumb et al. [2018] argued that point-fidelity can be misleading because it does not measure generalization of $e(x, f)$ across $N_x$, it has been used for evaluation in prior work [Ribeiro et al., 2016, 2018]. We report it in our experiments along with the neighborhood-fidelity for completeness.

[3]Each model takes less than a few minutes to train on an Intel 8700k CPU, so computational cost was not a limiting factor in our experiments. That being said, we observe a 2x speedup per iteration when using EXPO-1D-FIDELITY compared to EXPO-FIDELITY on the 'MSD' dataset and expect greater speedups on higher dimensional datasets.

[4]As in [Bloniarz et al., 2016], we treat the 'MSD' dataset as a regression problem with the goal of predicting the release year of a song.

[5]http://biostat.mc.vanderbilt.edu/wiki/Main/SupportDesc.

[6]Note that we do not ask participants to rate their trust in the model because of issues such as those raised in Lakkaraju and Bastani [2020].

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
