[Supplementary Material]

# A Appendix

## A.1 Comparison of EXPO to SENN

We compare against SENN [Alvarez-Melis and Jaakkola, 2018a] on the UCI 'breast cancer' dataset which is a binary classification problem. Because SENN's implementation outputs class probabilities, we run the post-hoc explainers on the probability output from the EXPO-regularized model as well (this differs from the 'support2' binary classification problem where we explain each logit individually). The results are in Table 5.

By comparing the first row, which shows the results for an EXPO-FIDELITY-regularized model whose regularization weight is tuned for accuracy, to the third row, which shows SENN's results, we can see that SENN's by-design approach to model interpretability seriously impaired its accuracy. However, SENN did produce a more interpretable model. From these results alone, there is no objective way to decide if the EXPO or SENN model is better. But, looking at the MAPLE-NF metric, we can see that its explanations have a standard error of around 4% relative to the model's predicted probability. This is reasonably small and probably acceptable for a model that makes a fraction as many mistakes.

Looking at the second row, which shows the results for a EXPO-FIDELITY-regularized model whose regularization weight has been increased to produce a model that is approximately accurate as SENN, we can see that the EXPO-regularized model is more interpretable than SENN.

Considering both of these results, we conclude that EXPO is more effective than SENN at improving the quality of post-hoc explanations.

However, this is a slightly unusual comparison because SENN is designed to produce its own explanations but we are using LIME/MAPLE to explain it. When we let SENN explain itself, it has a NF of 3.1e-5 and a Stability of 2.1e-3. These numbers are generally comparable to those of LIME explaining EXPO-Over Regularized. This further demonstrates EXPO's flexibility.

**Table 5:** A comparison of EXPO-FIDELITY, with the regularization weight tuned for accuracy and with it set too high to intentionally reduce accuracy, to SENN. Results are shown across 10 trials (with the standard error in parenthesis). EXPO can produce either a more accurate model or an equally accurate but more interpretable model.

| Method | Accuracy | MAPLE-PF | MAPLE-NF | MAPLE-S | LIME-PF | LIME-NF | LIME-S |
|---|---|---|---|---|---|---|---|
| EXPO | 0.99 (0.0034) | 0.013 (0.0065) | 0.039 (0.026) | 0.38 (0.27) | 0.1 (0.039) | 0.1 (0.039) | 0.0024 (0.0012) |
| EXPO- Over Regularized | 0.92 (0.013) | 0.00061 (0.00031) | 0.0014 (0.00079) | 0.0085 (0.01) | 0.0035 (0.0037) | 0.0035 (0.0037) | 2.4e-05 (1.2e-05) |
| SENN | 0.92 (0.033) | 0.0054 (0.0024) | 0.014 (0.0048) | 0.097 (0.044) | 0.012 (0.0043) | 0.013 (0.0045) | 0.00075 (0.00012) |

## A.2 Expanded Version of Section 2.2

This section provides the details for the results outlined in Section 2.2.

**Local explanations vs. Taylor approximations.** A natural question to ask is, *Why should we sample from $N_x$ in order to locally approximate $f$ when we could use the Taylor approximation as done in [Ross et al., 2017, Alvarez-Melis and Jaakkola, 2018a]?* The downside of a Taylor approximation-based approach is that such an approximation cannot be readily adjusted to different neighborhood scales and its fidelity and stability strictly depend on the learned function. Figure 3 shows that the Taylor approximations for two close points can be radically different from each other and are not necessarily faithful to the model outside of a small neighborhood.

**Fidelity regularization and the model's LC or TV.** From a theoretical perspective, EXPO-FIDELITY is similar to controlling the Lipschitz Constant or Total Variation of $f$ across $N_x$ after removing the part of $f$ explained by $e(x, f)$. From an interpretability perspective, having a large LC or TV does not necessarily lead to poor explanation quality, which is demonstrated in Figure 4.

**Figure 3:** A function (blue), its first order Taylor approximations at $x = 0.4$ (green) and $x = 0.5$ (red), and a local explanation of the function (orange) computed with $x = 0.5$ and $N_x = [0, 1]$. Notice that the Taylor approximation-based explanations are more strongly influenced by local variations in the function.

**Figure 4: Top:** Two functions (blue) and their local linear explanations (orange). The local explanations were computed with $x = 0.5$ and $N_x = [0, 1]$. **Bottom:** The unexplained portion of the function (residuals). **Comment:** Although both functions have a relatively large LC or TV, the one on the left is much better explained and this is reflected by its residuals.

**Standard Regularization.** We also consider two standard regularization techniques: $l_1$ and $l_2$ regularization. These regularizers may make the network simpler (due to sparser weights) or smoother, which may make it more amenable to local explanation. The results of this experiment are in Table 6; notice that neither of these regularizers had a significant effect on the interpretability metrics.

**Table 6:** Using $l_1$ or $l_2$ regularization has very little impact impact on the interpretability of the learned model.

| Metric | Regularizer | autompgs | communities | day | housing | music | winequality.red |
|---|---|---|---|---|---|---|---|
| MSE | None | 0.14 | 0.49 | 0.001 | 0.14 | 0.72 | 0.65 |
| | L1 | 0.12 | 0.46 | 1.7e-05 | 0.15 | 0.68 | 0.67 |
| | L2 | 0.13 | 0.47 | 0.00012 | 0.15 | 0.68 | 0.67 |
| MAPLE-PF | None | 0.016 | 0.16 | 0.001 | 0.057 | 0.17 | 0.013 |
| | L1 | 0.014 | 0.17 | 1.6e-05 | 0.054 | 0.17 | 0.015 |
| | L2 | 0.015 | 0.17 | 3.2e-05 | 0.05 | 0.17 | 0.02 |
| MAPLE-NF | None | 0.018 | 0.31 | 0.0012 | 0.066 | 0.18 | 0.013 |
| | L1 | 0.016 | 0.32 | 2.6e-05 | 0.065 | 0.18 | 0.016 |
| | L2 | 0.016 | 0.32 | 4.3e-05 | 0.058 | 0.17 | 0.021 |
| MAPLE-Stability | None | 0.015 | 1.2 | 2.6e-07 | 0.18 | 0.081 | 0.0043 |
| | L1 | 0.013 | 1.2 | 3e-07 | 0.21 | 0.072 | 0.004 |
| | L2 | 0.011 | 1.3 | 3.2e-06 | 0.17 | 0.065 | 0.0058 |
| LIME-PF | None | 0.04 | 0.1 | 0.0012 | 0.14 | 0.11 | 0.033 |
| | L1 | 0.035 | 0.12 | 0.00017 | 0.13 | 0.1 | 0.034 |
| | L2 | 0.037 | 0.12 | 0.00014 | 0.12 | 0.099 | 0.047 |
| LIME-NF | None | 0.041 | 0.11 | 0.0012 | 0.14 | 0.11 | 0.033 |
| | L1 | 0.036 | 0.12 | 0.00018 | 0.13 | 0.1 | 0.034 |
| | L2 | 0.037 | 0.12 | 0.00015 | 0.12 | 0.099 | 0.047 |
| LIME-Stability | None | 0.0011 | 0.022 | 0.00015 | 0.0047 | 0.011 | 0.0013 |
| | L1 | 0.0012 | 0.03 | 3e-05 | 0.0048 | 0.011 | 0.0016 |
| | L2 | 0.00097 | 0.032 | 1.7e-05 | 0.004 | 0.011 | 0.0021 |

## A.3 Model Details and Selection

The models we train are Multi-Layer Perceptrons with leaky-ReLU activations. The model architectures are chosen by a grid search over the possible widths (100, 200, 300, 400, or 500 units) and depths (1, 2, 3, 4, or 5 layers). The weights are initialized with the Xavier initialization and the biases are initialized to zero. The models are trained with SGD with the Adam optimizer and a learning rate of 0.001. For each dataset, the architecture with the best validation loss is chosen for final evaluation as the "None" model. Then, we use that same architecture and add the EXPO regularizer with weights chosen from 0.1, 0.05, 0.025, 0.01, 0.005, or 0.001. Note that these are not absolute weights and are instead relative weights: so picking 0.1 means that the absolute regularization weight is set such that the regularizer has $1/10^{th}$ the weight of the main loss function (estimated using a single mini-batch at the start of training and then never changed). This makes this hyper-parameter less sensitive to the model architecture, initialization, and dataset. We then pick the best regularization weight for each dataset using the validation loss and use that for the final evaluation as the EXPO model. Final evaluation is done by retraining the models using their chosen configurations and evaluating them on the test data.

## A.4 Defining the Local Neighborhood

**Choosing the neighborhood shape.** Defining a good regularization neighborhood, requires considering the following. On the one hand, we would like $N_x^{\text{reg}}$ to be similar to $N_x$, as used in Eq. 1 or Eq. 2, so that the neighborhoods used for regularization and for evaluation match. On the other hand, we would also like $N_x^{\text{reg}}$ to be consistent with the local neighborhood defined internally by $e$, which may differ from $N_x$. LIME can avoid this problem since the internal definition of the local neighborhood is a hyperparameter that we can set. However, for our experiments, we do not do this and, instead, use the default implementation. MAPLE cannot easily avoid this problem because the local neighborhood is learned from the data, and hence the regularization and explanation neighborhoods probably differ.

**Choosing $\sigma$ for $N_x$ and $N_x^{reg}$.** In Figure 5, we see that the choice of $\sigma$ for $N_x$ was not critical (the value of LIME-NF only increased slightly with $\sigma$) and that this choice of $\sigma$ for $N_x^{reg}$ produced slightly more interpretable models.

**Figure 5:** A comparison showing the effects of the $\sigma$ parameter of $N_x$ and $N_x^{reg}$ on the UCI Housing dataset. The LIME-NF metric grows slowly with $\sigma$ for $N_x$ as expected. Despite being very large, using $\sigma = 0.5$ for $N_x^{reg}$ is generally best for the LIME-NF metric.

## A.5 More Examples of EXPO's Effects

Here, we demonstrate the effects of EXPO-FIDELITY on more examples from the UCI 'housing' dataset (Table 7). Observe that the same general trends hold true:

- The explanation for the EXPO-regularized model more accurately reflects the model (LIME-NF metric)

- The explanation for the EXPO-regularized model generally considers fewer features to be relevant. We consider a feature to be 'significant' if its absolute value is 0.1 or greater.

- Neither model appears to be heavily influenced by CRIM or INDUS. The EXPO-regularized model generally relies more on LSTAT and less on DIS, RAD, and TAX to make its predictions.

**Table 7:** More examples comparing the explanations of a normally trained model ("None") to those of a EXPO-FIDELITY-regularized model. For each example we show: the feature values of the point being explained, the coefficients of the normal model's explanation, and the coefficients of the EXPO-regularized model's explanation. Note that the bias terms have been excluded from the explanations. We also report the LIME-NF metric of each explanation.

| Example Number | Value Shown | CRIM | INDUS | NOX | RM | AGE | DIS | RAD | TAX | PTRATIO | B | LSTAT | LIME-NF |
|---|---|---|---|---|---|---|---|---|---|---|---|---|---|
| 1 | $x$ | -0.36 | -0.57 | -0.86 | -1.11 | -0.14 | 0.95 | -0.74 | -1.02 | -0.22 | 0.46 | 0.53 | |
| | None | 0.01 | 0.03 | -0.14 | 0.31 | -0.1 | -0.29 | 0.27 | -0.26 | -0.07 | 0.13 | -0.24 | 0.0033 |
| | EXPO | 0.0 | 0.01 | -0.14 | 0.25 | 0.03 | -0.16 | 0.15 | -0.1 | -0.12 | -0.01 | -0.47 | 0.0033 |
| 2 | $x$ | -0.37 | -0.82 | -0.82 | 0.66 | -0.77 | 1.79 | -0.17 | -0.72 | 0.6 | 0.45 | -0.42 | |
| | None | 0.01 | 0.06 | -0.15 | 0.32 | -0.1 | -0.29 | 0.24 | -0.27 | -0.12 | 0.11 | -0.24 | 0.057 |
| | EXPO | 0.0 | 0.0 | -0.15 | 0.25 | 0.01 | -0.15 | 0.15 | -0.12 | -0.13 | 0.01 | -0.47 | 0.00076 |
| 3 | $x$ | -0.35 | -0.05 | -0.52 | -1.41 | 0.77 | -0.13 | -0.63 | -0.76 | 0.1 | 0.45 | 1.64 | |
| | None | -0.01 | 0.06 | -0.16 | 0.29 | -0.08 | -0.31 | 0.27 | -0.27 | -0.11 | 0.1 | -0.18 | 0.076 |
| | EXPO | -0.03 | -0.01 | -0.13 | 0.19 | -0.0 | -0.15 | 0.14 | -0.11 | -0.12 | 0.0 | -0.43 | 0.058 |
| 4 | $x$ | -0.36 | -0.34 | -0.26 | -0.29 | 0.73 | -0.56 | -0.51 | -0.12 | 1.14 | 0.44 | 0.14 | |
| | None | 0.02 | 0.06 | -0.18 | 0.29 | -0.1 | -0.34 | 0.31 | -0.21 | -0.09 | 0.12 | -0.27 | 0.10 |
| | EXPO | -0.02 | 0.01 | -0.13 | 0.21 | 0.02 | -0.16 | 0.17 | -0.11 | -0.12 | -0.0 | -0.47 | 0.013 |
| 5 | $x$ | -0.37 | -1.14 | -0.88 | 0.45 | -0.28 | -0.21 | -0.86 | -0.76 | -0.18 | 0.03 | -0.82 | |
| | None | 0.02 | 0.08 | -0.17 | 0.33 | -0.11 | -0.36 | 0.29 | -0.27 | -0.08 | 0.1 | -0.28 | 0.099 |
| | EXPO | -0.0 | -0.0 | -0.14 | 0.26 | 0.0 | -0.16 | 0.15 | -0.11 | -0.15 | 0.01 | -0.47 | 0.0021 |

The same comparison for examples from the UCI 'winequality-red' are in Table 8. We can see that the EXPO-regularized model depends more on 'volatile acidity' and less on 'sulphates' while usually agreeing about the effect of 'alcohol'. Further, it is better explained by those explanations than the normally trained model.

**Table 8:** The same setup as Table 7, but showing examples for the UCI 'winequality-red' dataset

| Example Number | Value Shown | fixed acidity | volatile acidity | citric acid | residual sugar | chlorides | free sulfur dioxide | total sulfur dioxide | density | pH | sulphates | alcohol | LIME-NF |
|---|---|---|---|---|---|---|---|---|---|---|---|---|---|
| 1 | $x$ | -0.28 | 1.55 | -1.31 | -0.02 | -0.26 | 3.12 | 1.35 | -0.25 | 0.41 | -0.2 | 0.29 | |
| | None | 0.02 | -0.11 | 0.14 | 0.08 | -0.1 | 0.05 | -0.15 | -0.13 | -0.01 | 0.31 | 0.29 | 0.021 |
| | EXPO | 0.08 | -0.22 | 0.01 | 0.04 | -0.04 | 0.06 | -0.12 | -0.09 | -0.01 | 0.17 | 0.3 | 6.6e-05 |
| 2 | $x$ | 1.86 | -1.91 | 1.22 | 0.87 | 0.39 | -1.1 | -0.69 | 1.48 | -0.22 | 1.96 | -0.35 | |
| | None | 0.02 | -0.15 | 0.11 | 0.07 | -0.08 | 0.07 | -0.23 | -0.09 | -0.06 | 0.3 | 0.27 | 0.033 |
| | EXPO | 0.09 | -0.23 | 0.02 | 0.04 | -0.05 | 0.06 | -0.13 | -0.09 | -0.0 | 0.18 | 0.3 | 0.0026 |
| 3 | $x$ | -0.63 | -0.82 | 0.56 | 0.11 | -0.39 | 0.72 | -0.11 | -1.59 | 0.16 | 0.42 | 2.21 | |
| | None | 0.03 | -0.1 | 0.13 | 0.05 | -0.06 | 0.12 | -0.21 | -0.19 | -0.08 | 0.38 | 0.29 | 0.11 |
| | EXPO | 0.09 | -0.22 | 0.02 | 0.04 | -0.04 | 0.06 | -0.12 | -0.09 | -0.0 | 0.18 | 0.3 | 8.2e-05 |
| 4 | $x$ | -0.51 | -0.66 | -0.15 | -0.53 | -0.43 | 0.24 | 0.04 | -0.56 | 0.35 | -0.2 | -0.07 | |
| | None | 0.03 | -0.16 | 0.12 | 0.05 | -0.13 | 0.09 | -0.21 | -0.13 | -0.05 | 0.35 | 0.24 | 0.61 |
| | EXPO | 0.09 | -0.22 | 0.01 | 0.04 | -0.04 | 0.06 | -0.12 | -0.09 | -0.0 | 0.18 | 0.3 | 6.8e-05 |
| 5 | $x$ | -0.28 | 0.43 | 0.1 | -0.65 | 0.61 | -0.62 | -0.51 | 0.36 | -0.35 | 5.6 | -1.26 | |
| | None | 0.03 | -0.12 | 0.09 | 0.12 | -0.11 | 0.03 | -0.19 | -0.13 | -0.03 | 0.13 | 0.24 | 0.19 |
| | EXPO | 0.08 | -0.22 | 0.02 | 0.04 | -0.05 | 0.05 | -0.13 | -0.09 | -0.0 | 0.16 | 0.3 | 0.0082 |

## A.6 Quantitative Results on the 'support2' Dataset

In Table 9, we compare EXPO-regularized models to normally trained models on the 'support2' dataset.

**Table 9:** A normally trained model ("None") vs. the same model trained with EXPO-FIDELITY or EXPO-1D-FIDELITY on the 'support2' binary classification dataset. Each explanation metric was computed for both the positive and the negative class logits. Results are shown across 10 trials (with the standard error in parenthesis). Improvement due to FIDELITY and 1D-FIDELITY over the normally trained model is statistically significant ($p = 0.05$, t-test) for all of the metrics.

| Output | Regularizer | LIME-PF | LIME-NF | LIME-S | MAPLE-PF | MAPLE-NF | MAPLE-S |
|--------|-------------|---------|---------|--------|----------|----------|---------|
| Positive | None | 0.177 (0.063) | 0.182 (0.065) | 0.0255 (0.0084) | 0.024 (0.008) | 0.035 (0.010) | 0.34 (0.06) |
| | FIDELITY | **0.050 (0.008)** | **0.051 (0.008)** | **0.0047 (0.0008)** | **0.013 (0.004)** | **0.018 (0.005)** | **0.13 (0.05)** |
| | 1D-FIDELITY | 0.082 (0.025) | 0.085 (0.025) | 0.0076 (0.0022) | 0.019 (0.005) | 0.025 (0.005) | 0.16 (0.03) |
| Negative | None | 0.198 (0.078) | 0.205 (0.080) | 0.0289 (0.0121) | 0.028 (0.010) | 0.040 (0.014) | 0.37 (0.18) |
| | FIDELITY | **0.050 (0.008)** | **0.051 (0.008)** | **0.0047 (0.0008)** | **0.013 (0.004)** | **0.018 (0.005)** | **0.13 (0.03)** |
| | 1D-FIDELITY | 0.081 (0.026) | 0.082 (0.027) | 0.0073 (0.0021) | 0.019 (0.006) | 0.024 (0.007) | 0.16 (0.06) |

**Accuracy (%):** None: $83.0 \pm 0.3$, FIDELITY: $83.4 \pm 0.4$, 1D-FIDELITY: $82.0 \pm 0.3$.

### A.7   User Study: Additional Details

**Data Details.** Figure 6 shows the histogram of the number of steps participants take to complete each round. We use 150 as our cut-off value for removing participant's data from the final evaluation. Figure 7 shows a histogram of the number of steps participants take to complete each round. There is no evidence to suggest that the earlier rounds took a different amount of time than the later rounds. So learning effects were not significant in this data.

Figure 7: A series of histograms showing how many steps participants take to complete each round for each condition. Generally, there is no evidence of learning effects over the course of the five rounds.

**Figure 6:** A histogram showing the number of steps participants take to complete each round.

**Algorithmic Agent.** In addition to measuring humans' performance on this task (see Section 5 for details), we are also interested in measuring a simple algorithmic agent's performance on it. The benefit of this evaluation is that the agent relies solely on the information given in the explanations and, as a result, does not experience any learning affects that could confound our results.

Intuitively, we could define a simple greedy agent by having it change the feature whose estimated effect is closest to the target change. However, this heuristic can lead to loops that the agent will never escape. As a result, we consider a randomized version of this greedy agent.

Let $\lambda$ the degree of randomization for the agent, $y$ denote the model's current prediction, $t$ denote the target value, and $c_i$ denote the explanation coefficient of feature $i$. Then the score of feature $i$, which measures how close this features estimated effect is to the target change, is: $s_i = -\lambda * ||c_i| - |y - t||$. The agent then chooses to use feature $i$ with probability $\frac{e^{s_i}}{\sum_j e^{s_j}}$.

Looking at this distribution, we see that it is uniform (*i.e.*, does not use the explanation at all) when $\lambda = 0$ and that it approaches the greedy agent as $\lambda$ approaches infinity.

In Figure 8, we run a search across the value of $\lambda$ to find a rough trade-off between more frequently using the information in the explanation and avoiding loops. Note that the agent performs better for the EXPO-regularized model.

**Figure 8:** A comparison of the average number of steps it takes for either a human or an algorithmic agent to complete our task. The x-axis is a measure of the agent's randomness: 0 corresponds to a totally random agent with increasing values indicating a greedier agent. Both humans and the agent find it easier to complete the task for the EXPO-regularized model.

## A.8 Non-Semantic Features

When $\mathcal{X}$ consists of non-semantic features, we cannot assign meaning to the difference between $x$ and $x'$ as we could with semantic features. Hence it does not make sense to explain the difference between the predictions $f(x)$ and $f(x')$ and fidelity is not an appropriate metric.

Instead, for non-semantic features, local explanations try to identify which parts of the input are particularly influential on a prediction [Lundberg and Lee, 2017, Sundararajan et al., 2017]. Consequently, we consider explanations of the form $\mathcal{E}_{ns} := \mathbb{R}^d$, where $d$ is the number of features in $\mathcal{X}$, and our primary explanation metric is stability. Note that we could consider these types of local explanations for semantic features as well, but that they answer a different question than the approximation-based local explanations we consider.

**Post-hoc explainers.** Various explainers [Sundararajan et al., 2017, Zeiler and Fergus, 2014, Shrikumar et al., 2016, Smilkov et al., 2017] have been proposed to generate local explanations in $\mathcal{E}_{ns}$ for images. However, it should be noted that the usefulness and evaluation of these methods is uncertain [Adebayo et al., 2018, Tomsett et al., 2019]. For our experiment, we will consider *saliency maps* [Simonyan et al., 2013] which assign importance weights to image pixels based on the magnitude of the gradient of the predicted class with respect to the corresponding pixels.

**Stability Regularizer.** For EXPO-STABILITY, we simply require that the model's output not change too much across $N_x^{reg}$ (Algorithm 2). A similar procedure was explored previously in [Zheng et al., 2016] for adversarial robustness.

**Experimental setup.** For this experiment, we compared a normally trained convolutional neural network on MNIST to one trained using EXPO-STABILITY. Then, we evaluated the quality of saliency map explanations for these models. Both $N_x$ and $N_x^{reg}$ where defined as $\mathrm{Unif}(x - 0.05, x + 0.05)$. Both the normally trained model and model trained with EXPO-STABILITY achieved the same accuracy of 99%. Quantitatively, training the model with EXPO-STABILITY decreased the stability metric from 6.94 to 0.0008. Qualitatively, training the model with EXPO-STABILITY made the resulting saliency maps look much better by focusing them on the presence or absence of certain pen strokes (Figure 9).

---

**Algorithm 2** Neighborhood-stability regularizer

---

**input** $f_\theta, x, N_x^{\mathrm{reg}}, m$
1: Sample points: $x'_1, \ldots, x'_m \sim N_x^{\mathrm{reg}}$
2: Compute predictions:

$$\hat{y}_j(\theta) = f_\theta(x'_j), \text{ for } j = 1, \ldots, m$$

**output** $\frac{1}{m} \sum_{j=1}^{m} (\hat{y}_j(\theta) - f(x))^2$

---

**Figure 9:** Original images (left) and saliency maps of an normally trained model (middle) and a EXPO-STABILITY-regularized model (right).