[Reviews · NeurIPS 2020]

Review 1

Summary and Contributions: This paper proposes a method (EXPO) to improve interpretability of models trained through gradient descent through regularization. They propose two regularizers --- one that increases the fidelity of explanations (i.e. local accuracy of model within neighborhood) and stability (i.e. are points uniformly classified within neighborhood). Effectively, the fidelity and stability regularizers encourage models to behave linearly and uniformly respectively. They find that their regularizers lead to improved local explainability as measured through local fidelity. Also, they perform comprehensive users studies. They ask users to determine what they need to change about housing prices in order to score positive outcomes using an explanation produced through a regularized model and an unregularized model. They find users perform significantly better using the regularized models. Lastly, they demonstrate improved quality of saliency maps and superiority as compared to l1 and l2 regularization.

Strengths: Encouraging models to behave locally linear is an interesting idea. The results demonstrated are compelling and the proposed regularizers do seem to lead to positive local effects for explainability. Also, the regularizers seem easy to implement and could offer straight forward adoption for practitioners --- the simplicity here is taken in a positive way. The proposed methods seem relevant to explainability at NeurIPS and offer a new approach to train models that are more interpretable. Further, the user study evaluation provides an interesting and particularly nice evaluation dimension. Asking users to actually make changes based on their received explanations and then gauging how successful they are seems like a realistic approach to evaluation and results given are compelling. Last, I haven't seen a study quite like this for assessing feature importance explainability methods. I think this could present an interesting new dimension for researchers to assess explainability methods.

Weaknesses: I was confused about some of the experimental details. The authors mention they compare "normally" trained models to models regularized through EXPO. Perhaps I'm missing something, but what exactly are these models? They mention the network architecture is chosen through grid search indicating that they are using neural networks. Looking at the code, I can confirm an MLP is being used. However, providing these details clearly in the paper and appendix would be a plus. This felt like a bit of a gap in the paper currently to me, but please correct me if I'm not seeing something. I appreciate that code is shared, but currently I think it would be difficult to reproduce this work from the paper alone. My main concern with the paper derives from the significance of the work. The idea of regularizing for local linearity feels related to work in the adversarial robustness literature. In particular, one paper comes to mind "Adversarial Robustness through Local Linearization" by Qin et. al. where they propose regularizing for local linearity to encourage robustness to adversarial attacks. Given the simplicity of the approach combined with the potential lack of novelty, I'm somewhat worried about the significance -- though I understand previous work is taken in a different direction. --- Update: Thanks for the responses. It will be useful to have hyperparameter details available in the main paper, so I've updated my reproducibility scores and my overall score. One minor comment still is that I would recommend staying away from language like "normally trained models" and instead say something like "Models trained without regularization." I found the language around normally trained models confusing while reading.

Correctness: The claims and methods seem correct. Because one of the major contributions of this paper is a user study, preregistered hypotheses and evaluation would have been a plus.

Clarity: The paper is well written and easy to read.

Relation to Prior Work: The authors include a nice section differentiating their work from other related regularization explainability work. As, I mentioned earlier, it could be relevant to discuss other methods that regularize for local linearity.

Reproducibility: Yes

Additional Feedback: Though I have some concerns with the overall novelty of the work, I am inclined to recommend acceptance given the ease of adoption of the regularizers and compelling user study evidence. I would be willing to raise my score if more experimental details are included in the paper. I currently don't feel like I could reproduce the paper as it stands, but please let me know if am just not seeing details related to model architecture, learning rate, etc.


Review 2

Summary and Contributions: The paper proposes regularizing machine learning model for better interpretability. In particular, they bridge post-explanation and during-training explanation approaches by adding a regularization term for explainability to the loss function. Better explainability is here defined as either a more stable explanation around a point x or being close to the output of an interpretable model. For the fidelity version this means that the model is penalized to be linearly dependent on the input in the neighbourhood of x (with x being a given input). *** I have read the rebuttal. My score remains unchanged.

Strengths: The paper is well-written, easy to read and clearly motivates its claim. The claim (models regularized with EXPO are better to interpret) is validated on a number of datasets. The authors also do a solid usability study. As far as I am aware, the idea of using explanation regularization to improve explanations without prior knowledge is novel.

Weaknesses: As the computational complexity scales with the cube of the number of dimensions, EXPO is not usable for higher-dimensional inputs. This is also reflected in the datasets chosen for validation, with all of them being relatively low-dimensional. Considering that explainability is especially of interest for complex models with a high-dimensional input, this severely limits the significance of the method and should also be more clearly outlined and the paper. There is also already previous literature on regularizing explanations with prior knowledge where “Explanations should be locally consistent” could be used as prior knowledge. The authors could f.e. compare to RRR in this manner.

Correctness: To the best of my knowledge, the claims and methods are correct and applied correctly.

Clarity: Largely, the paper is well written. Section 2.2 is confusing and the purpose of it is not apparent to me. A claim like "Finally, we empirically show that standard l1 or l2 regularization techniques do not influence model interpretability." would imply to me that I am about to read the empirical proof that this is the case.

Relation to Prior Work: It is clearly described how the paper differs from previous work. The authors should do a more thorough literature review on regularizing explanations, f.e. Du, Mengnan, et al. "Learning credible deep neural networks with rationale regularization." 2019 IEEE International Conference on Data Mining (ICDM). IEEE, 2019. Weinberger, Ethan, Joseph Janizek, and Su-In Lee. "Learning Deep Attribution Priors Based On Prior Knowledge." arXiv (2019): arXiv-1912. Laura Rieger, Chandan Singh, William Murdoch, and Bin Yu. Interpretations are useful: Penalizing explanations to align neural networks with prior knowledge. In Proceedings of Machine Learning and Systems 2020, pages 1598–1608. 2020.

Reproducibility: Yes

Additional Feedback: The architectures used should be described in more detail than "chosen using a grid search". Minor nitpick but I feel that empirical results and the user study would not be contributions in l. 47-56. In Fig 1., the legend for ExpO should be more clearly marked as a legend.


Review 3

Summary and Contributions: The authors propose to regularize black-box models using "whole neighborhood" versions of one of two existing interpretability regularizer, one focusing on fidelity and one focusing on stability. The aim is that models regularized in this fashion allow post-hoc explainers (like LIME) to extract more faithful and well-behaved explanations. Post-rebuttal update: My score gave too little credit to the user study, so I increased it a bit. I still think that technical novelty is limited and that the authors should be very clear about this; as I had written, this could easily be solved by focusing the paper on the user study and on the empirical results, as in my opinion the message does *not* depend on technical novelty at all. I strongly encourage the authors to rework the text accordingly.

Strengths: - Contribution is very intuitive. - Reasonably well written. - User study. I enjoyed reading it, but I am not a behavioral psychologist and cannot really provide feedback about it (e.g. I cannot spot obvious biases in the experiment as I have no training to do so).

Weaknesses: - Limited novelty. Novelty: The overall idea of making LIME's job easier makes sense, and it is the one key contribution of this paper. However, the way to get there is lifted directly from previous work. The difference between RRR, SENN, etc. is that a neighborhood (which must be estimated) is introduced. This is fine, but not a huge difference conceptually. Algorithm 1 computes the residual of the best linear fit in a neighborhood (notice that it can be rewritten in a single line), so it's not very novel. Algorithm 2 (for some reason reported only in the supplement) computes variations in prediction output, also not particularly novel. The idea of penalizing deviations from linearity etc. is not too dissimilar to previous work. More advanced regularizers (there are a couple of papers on tree regularization mentioned somewhere else in the review) are available which aim at achieving a similar effect as EXPO except for more general classes of explanations. All in all, I am afraid that the novelty of the approach is not too obvious. I suggest to focus the message more on the usage/consequences of their technique than on the technique itself -- as it is not too impressive. The empirical results, and especially the user study, are definitely more interesting.

Correctness: The empirical results show that EXPO improves the all the metrics it sets to improve. This is not very surprising, as the overall idea is to *optimize* those metrics during learning in a very straightforward fashion. But it seems that the method and evaluation are correct. My experience with LIME is that it has terrible variance, depending on the kind of variables (e.g. categorical) and kernel width. I am very surprised that LIME (in no-EXPO, "None") case works so well in terms of variance. I also do not understand why the accuracy of SENN explanations should be measured using a post-hoc explainers. SENNs are self-explainable, they are designed *not* to require a post-hoc explainer at all. Nobody should apply LIME to SENNs. The accuracy of SENN explanations w.r.t. the SENN model is always one, by definition. SENNs have no notion of neighborhood, so it is not clear why it should be applied to them.

Clarity: The text is well written and easy to follow. Ideas are stated clearly. A couple of remarks: - Some contributions are hidden in the appendices. For instance, the algorithm for computing EXPO-STABILITY appears in the very last section of the supplementary material -- where the authors mention that it does the same as prior work (where it was intended as an approach for robustifying the model). - It is not clear to me why N_x is a probability distribution (line 60).

Relation to Prior Work: The related work is mostly complete. A couple of missing relevant papers are: Beyond sparsity: Tree regularization of deep models for interpretability, AAAI'18 and its follow-up: Regional Tree Regularization for Interpretability in Deep Neural Networks, AAAI'20. These approaches regularize neural nets (or other black-box models) so that they behave similarly to decision trees, either globally or regionally. I would expect tree-regularized models to work well together with LIME and other post-hoc local explainers -- and, notably, to work well for both linear explanations and tree-based explanations.

Reproducibility: Yes

Additional Feedback: Broader impact: The authors do not consider the effect of expo on "explanation fooling" (see for instance "Fooling lime and shap: Adversarial attacks on post hoc explanation methods", 2020; or Explanations can be manipulated and geometry is to blame, 2019) and, perhaps more importantly, the fact that local explanations can be used to persuade/manipulate humans into believing that a model performs well everywhere, which may well be not the case. To be clear, my main issue with the paper is the lack of novelty. Also, I am willing to increase my score if the authors address my concerns.


Review 4

Summary and Contributions: This paper addresses the problem of the interpretability of black-box models using regularization. Two regularizers are proposed to derive the fidelity and stability for post-hoc explanation. The paper evaluates the proposed approaches on multiple datasets. A user study is included to demonstrate the improvement in fidelity and stability.

Strengths: 1. This paper addresses an important problem of model interpretability. The proposed solution is model-agnostic, which can be applied to many machine learning models. 2. It is interesting to investigate human-grounded metrics in model interpretability.

Weaknesses: 1. The novelty of this paper is incremental. The paper merely combines the post-hoc explainers with fidelity metric and stability metric. 2. The experimental part is somewhat not convincing. * The experiments use the fidelity and stability metric to measure the proposed approach, which is optimized for those two metrics. Therefore, the metrics might not be sufficient to measure the proposed approach. Other measurements for model interpretability should be included. * The experimental results show that the proposed approach with regularizers outperforms the normally trained model without regularizers. This result is straightforward. Also, it is not surprising to see the results in user study: the regularized model achieves better interpretability than the normal model. The proposed approach should be compared with other interpreting methods in user study. 3. Minor comment: e is not defined in Equation 2.

Correctness: The method is reasonable. The empirical analysis is not convincing.

Clarity: The paper is well-written and easy to read.

Relation to Prior Work: Yes

Reproducibility: Yes

Additional Feedback:

[Author Response · NeurIPS 2020]

Thank you for your thoughtful feedback. We will first discuss common themes and then specific reviewer comments.

**Significance:** Even though ExpO is "simple" (in that it connects existing concepts, albeit in a novel way), we believe
that it is highly impactful because there is no other model-agnostic and domain-knowledge free method for improving
the quality of local approximation explanations such as LIME (which is a seminal method in Interpretable ML).

**Prior Work:** The suggested related works (which we will cite in the revision) all solve different problems than the one
we consider. We will add a discussion as outlined below.

• "Adversarial Robustness ..." by Qin et al does not consider interpretability at all. When adapted to consider
interpretability, it uses a gradient based explanation and its regularizer is quite similar to SENN's. Consequently, it
will have the same issues with flexibility, fidelity, and stability as gradient based explanations. See A.2 for details.

• Several methods rely on domain knowledge: "Learning credible ..." by Du et al, "Learning Deep ..." by Weinberger
et al, "Interpretations are ..." by Rieger et al, and "Regional Tree ..." by Wu et al [2].

• "Beyond sparsity ..." by Wu et al [1] regularizes for global interpretability while ExpO regularizes for local
interpretability. Despite the fact that they are globally interpretable, small decision trees are difficult to explain locally
with explainers like LIME (see Figure 1 for an example). As a result, [1,2] do not solve the same problem as ExpO
because making the model look more like a decision tree makes LIME less effective.

**Reviewer 1.** *Reproducibility.* The reviewer is correct that we are comparing MLPs trained using standard techniques to
ones trained with ExpO. We will add a detailed discussion of the neural networks (structure, activations, widths, depths,
etc), hyper-parameters (learning rate, optimizer, regularization), and selection procedures to the appendix so that the
reader does not have to reference the code (which reproduces all of our results) to reproduce our results.

**Reviewer 2.**

*"computational complexity ... cube ... not usable for higher-dimensional inputs."* We introduce ExpO-1D-Fidelity to
address this concern (line 160-167). Its complexity is independent of the data dimension and we show it scales well to
datasets with $\sim 100$ features. We also note that related methods require expensive operations (FTSD and [1] both are
non-differentiable; SENN and RRR both require differentiating through the model gradient).

*"compare to RRR in this manner."* To the best of our knowledge, it is not technically possible to encode fidelity/stability
using RRR's regularizer.

**Reviewer 3.**

*"difference between RRR, SENN is that a neighborhood ... is introduced ... not a huge difference."* ExpO is the only
method that is differentiable and model agnostic that does not require domain knowledge; the differences are not just in
whether or not a neighborhood is used. See Table 1 for details.

*"Algorithm 1 ... not very novel."* Viewing the novelty of ExpO merely through the lens of Algorithm 1 sells it short; the
novelty stems from its impactful connection to interpretability. It is common for algorithms designed in one area to be
impactful when introduced to another area (eg, SENN/RRR are "just" regularizing the gradient which is a strategy at
least as old as "Tangent prop-a formalism for specifying selected invariances in an adaptive network." NeurIPS92.)

*"results ... not very surprising ... idea is to \*optimize\* those metrics during learning."* Two small clarifications: the
results are shown for points that were not regularized for during training and the results shown in the main paper were
regularized only for fidelity, so the improvement in stability is not a given.

*"why is the accuracy of SENN explanations...measured using a post-hoc explainers."* While the reviewer is correct
that SENN's Point-Fidelity (PF) is perfect by-design, its Neighborhood-Fidelity (NF) is not guaranteed; the setup of
user study clearly motivates why NF can be preferable to PF (lines 220 - 223). Following the reviewer's suggestion,
we computed NF and Stability for SENN explaining itself. While the results are better than using LIME, they
corroborate the general message that ExpO is a more flexible solution than SENN for trading off between accuracy and
interpretability. Specifically, SENN explaining itself has a NF of 3.1e-5 and a Stability of 2.1e-3; these numbers are
generally comparable to LIME explaining the appropriate ExpO model. See A.1 and Table 5 for details.

*"regularize neural nets ... behave similarly to decision trees, either globally or regionally ... expect tree-regularized
models to work well together with LIME...for both linear explanations and tree-based explanations."* As noted in the
above discussion on [1], neither of these methods would improve LIME's explanation quality for linear explanations.
Although we agree that exploring non-linear local explanations is an interesting direction, ExpO focuses on the setting
where the explanation is linear because this is what LIME, MAPLE, and SENN all do.

**Reviewer 4:** *"experimental part is somewhat not convincing ... it is not surprising to see the results in user study: the
regularized model achieves better interpretability than the normal model."* Fidelity/stability are the standard proxy
metrics used to evaluate local approximations. However, as we emphasize in the paper (line 38-42), they are only
proxies for some underlying notion of interpretability, and the goal of the user study is to directly study explanation
usefulness. Consequently, it inconsistent to criticize the results on the metrics and then use those same results to criticize
the results of the user study.

[Meta-Review · NeurIPS 2020]

The reviewers agree that the problem paper is trying to solve is quite important, and through very strong experiments, show that the approach is quite effective. The biggest concern that was brought up in the discussion was the lack of novelty. The reviewers strongly urge the authors to focus more on the strength of their contributions, in particular the evaluation, as opposed to trying to differentiate from work they are clearly building upon.